# Protective Effects of Nargenicin A1 against Tacrolimus-Induced Oxidative Stress in Hirame Natural Embryo Cells

**DOI:** 10.3390/ijerph16061044

**Published:** 2019-03-22

**Authors:** Cheol Park, Da Hye Kwon, Su Jung Hwang, Min Ho Han, Jin-Woo Jeong, Sang Hoon Hong, Hee-Jae Cha, Su-Hyun Hong, Gi-Young Kim, Hyo-Jong Lee, Suhkmann Kim, Heui-Soo Kim, Yung Hyun Choi

**Affiliations:** 1Department of Molecular Biology, College of Natural Sciences, Dong-eui University, Busan 47340, Korea; parkch@deu.ac.kr; 2Department of Biochemistry, Dong-eui University College of Korean Medicine, Busan 47227, Korea; chghl1013@hanmail.net (D.H.K.); hongsh@deu.ac.kr (S.-H.H.); 3Department of Pharmacy, College of Pharmacy, Inje University, Gimhae 50834, Korea; sama3575@naver.com (S.J.H.); hjlee@inje.ac.kr (H.-J.L.); 4National Marine Biodiversity Institute of Korea, Seocheon 33662, Korea; mhhan@mabik.re.kr; 5Nakdonggang National Institute of Biological Resources, Sangju 17104, Korea; jinwooyo@nate.com; 6Department of Internal Medicine, Dong-eui University College of Korean Medicine, Busan 47227, Korea; shhong@deu.ac.kr; 7Department of Parasitology and Genetics, Kosin University College of Medicine, Busan 49267, Korea; hcha@kosin.ac.kr; 8Department of Marine Life Sciences, Jeju National University, Jeju 63243, Korea; immunkim@jejunu.ac.kr; 9Department of Chemistry, College of Natural Sciences, Center for Proteome Biophysics and Chemistry Institute for Functional Materials, Pusan National University, Busan 46241, Korea; suhkmann@gmail.com; 10Department of Biological Sciences, College of Natural Sciences, Pusan National University, Busan 46241, Korea; khs307@pusan.ac.kr

**Keywords:** nargenicin A1, tacrolimus, ROS, DNA damage, apoptosis

## Abstract

Tacrolimus is widely used as an immunosuppressant to reduce the risk of rejection after organ transplantation, but its cytotoxicity is problematic. Nargenicin A1 is an antibiotic extracted from *Nocardia argentinensis* and is known to have antioxidant activity, though its mode of action is unknown. The present study was undertaken to evaluate the protective effects of nargenicin A1 on DNA damage and apoptosis induced by tacrolimus in hirame natural embryo (HINAE) cells. We found that reduced HINAE cell survival by tacrolimus was due to the induction of DNA damage and apoptosis, both of which were prevented by co-treating nargenicin A1 or N-acetyl-l-cysteine, a reactive oxygen species (ROS) scavenger, with tacrolimus. In addition, apoptosis induction by tacrolimus was accompanied by increases in ROS generation and decreases in adenosine triphosphate (ATP) levels caused by mitochondrial dysfunction, and these changes were significantly attenuated in the presence of nargenicin A1, which further indicated tacrolimus-induced apoptosis involved an oxidative stress-associated mechanism. Furthermore, nargenicin A1 suppressed tacrolimus-induced B-cell lymphoma-2 (Bcl-2) down-regulation, Bax up-regulation, and caspase-3 activation. Collectively, these results demonstrate that nargenicin A1 protects HINAE cells against tacrolimus-induced DNA damage and apoptosis, at least in part, by scavenging ROS and thus suppressing the mitochondrial-dependent apoptotic pathway.

## 1. Introduction

Tacrolimus, also known as FK506, is a macrolide antibiotic isolated from the fermentation products of the soil fungus *Streptomyces tsukubaensis* [1]. It is a potent immunosuppressive that inhibits T cell activation, T helper cell-mediated B-cell proliferation, and cytokine formation by disrupting calcineurin-mediated signaling in the same manner as cyclosporin A. However, the efficacy of immunosuppression is much more potent than that of cyclosporin A [2,3]. Due to its immunosuppressive effects, tacrolimus is used clinically to manage immune rejection after solid organ transplantation and to treat autoimmune diseases [3,4]. In addition, tacrolimus has been reported to protect cells from apoptotic and necrotic cell death and to have neuroprotective and neuroregenerative effects due to its suppression of proinflammatory cytokine levels [5,6].

However, long-term use of tacrolimus has been reported to affect organ transplant survival adversely because its chronic administration has been associated with nephrotoxicity, diabetes, neurotoxicity, and gastrointestinal disturbances [7,8,9]. In this regard, several studies have shown tacrolimus is toxic to renal proximal tubular epithelial cells, insulin-secreting cells, and gastric and lung epithelial cells [10,11,12]. Furthermore, the excessive production of inflammatory regulators such as cyclooxygenase-2 and transforming growth factor-β has been reported to promote tacrolimus-induced glomerular and tubular cell damage [13,14]. Notably, the overproduction of reactive oxygen species (ROS), byproducts of aerobic respiration, and diminished adenosine triphosphate (ATP) production associated with impaired mitochondrial function have been proposed to be major causes of tacrolimus cytotoxicity [15,16]. Thus, it would appear blocking oxidative stress and maintaining energy homeostasis provide a potential means of reducing the cytotoxicity of tacrolimus.

Although ROS act as signaling molecules and are essential for cell growth and proliferation, persistently high intracellular ROS levels can cause oxidative damage [17,18]. Mitochondria are major sources of ROS and its most vulnerable targets, and excessive ROS accumulation is considered to be a major cause of DNA damage-mediated apoptosis [19,20]. Intracellular ROS accumulation beyond the antioxidant capacities of cells also reduces mitochondrial membrane potentials (MMPs) and compromises ATP production. And resultantly, apoptogenic factors including cytochrome *c* are released into cytoplasm from the mitochondrial intermembrane space and activate the caspase cascade leading to apoptosis [19,21].

Nargenicin A1 is a major secondary metabolite isolated from cultures of *Nocardia argentinensis*. It is a natural antibiotic, with strong antibacterial activity against various Gram-positive bacteria including methicillin-resistant *Staphylococcus aureus* [22,23]. Besides its antibacterial activity, nargenicin A1 has been shown to inhibit leukemia cell proliferation and promote leukemia cell differentiation, thus being useful for the treatment of neoplastic diseases [24]. This compound has also been suggested to be evaluated as a therapeutic agent for inflammatory neurodegenerative diseases by significantly attenuating the lipopolysaccharide-induced inflammatory response in microglia [25]. In addition, nargenicin has been reported to have antioxidant efficacy [26], but molecular events responsible for its activity have not yet been determined. The present study was undertaken to evaluate the protective effects of nargenicin A1 on DNA damage and apoptosis induced by tacrolimus in hirame natural embryo (HINAE) cells, and was conducted as a part of a study aimed at identifying agents that protect against the adverse effects of tacrolimus.

## 2. Materials and Methods

### 2.1. Cell Culture and Drug Treatment

The HINAE cell line, which was derived from Japanese flounder embryos [27], was provided by Dr. Jaehun Cheong (Department of Molecular Biology, College of Natural Sciences, Pusan National University, Busan, Republic of Korea). Cells were cultured in Leibovitz’s L-15 medium (Life Technologies, Carlsbad, CA, USA) containing 10% fetal bovine serum (FBS, WelGENE Inc., Daegu, Republic of Korea), 100 U/mL penicillin, and 100 U/mL streptomycin (WelGENE Inc.) at 20 °C. Tacrolimus and nargenicin A1 were purchased from Sigma-Aldrich Chemical Co. (St. Louis, MO, USA) and Abcam (Cambridge, UK), respectively, and dissolved in dimethyl sulfoxide (DMSO, Sigma-Aldrich Chemical Co.). The final concentration of DMSO did not exceed 0.05%, and did not show cytotoxicity. To treat HINAE cells, each stock solution (tacrolimus 10 mM, nargenicin A1 1 mM) was diluted in complete culture medium as appropriate to achieve final concentrations.

### 2.2. Cell Viability Assay

For the cell viability study, 5 × 10^3^ HINAE cells were seeded per well in 96-well plates, incubated for 24 h, and then incubated with different concentrations of tacrolimus or nargenicin A1 for 24 h or pre-incubated with nargenicin A1 for 1 h before being treated with tacrolimus for 24 h. Cells were also treated with *N*-acetyl cysteine (NAC, Sigma-Aldrich Chemical Co.), a known ROS scavenger, for 1 h in the presence or absence of tacrolimus. Subsequently, cell viability was determined using an 3-(4,5-dimethylthiazol-2-yl)-2,5-diphenyltetrazolium bromide (MTT; Sigma-Aldrich Chemical Co.) assay as previously described [28]. The experiments were repeated three times with at least triplicate wells for each concentration, and results are expressed as percentage reductions in absorbance versus non-treated controls.

### 2.3. Apoptosis Assay Using a Fluorescence Microscope

Apoptosis induction was evaluated by observing nuclear morphological changes. Briefly, HINAE cells were seeded on 6-well plates at a density of 3 × 10^5^ cells/well for 24 h and treated with or without nargenicin A1 or NAC for 1 h before adding tacrolimus for a further 24 h. Cells were then harvested, washed with phosphate buffered saline (PBS), and fixed with 4% paraformaldehyde (Sigma-Aldrich Chemical Co.) in PBS for 30 min at room temperature. Cells were then permeabilized with 0.1% (*w*/*v*) Triton X-100 for 5 min, stained with 1.0 mg/mL of 4,6-diamidino-2-phenylindole (DAPI, Sigma-Aldrich Chemical Co.) solution for 10 min at room temperature in the dark, and washed twice with PBS. Nuclear morphologic changes were examined using a fluorescence microscope (Carl Zeiss, Oberkochen, Germany) at × 400 magnification.

### 2.4. Apoptosis Analysis Using a Flow Cytometer

Apoptosis extents were determined by flow cytometry using the Annexin V/propidium iodide (PI) double staining. After treating cells with tacrolimus with or without nargenicin A1 or NAC, cells were collected, rinsed with PBS, re-suspended in binding buffer, and stained with fluorescein isothiocyanate (FITC)-conjugated annexin V and PI (BD Pharmingen, San Diego, CA, USA) at room temperature for 20 min in the dark, according to the manufacturer’s instructions. Cell fluorescence intensities were detected using a flow cytometer (Becton Dickinson, San Jose, CA, USA) and data were analyzed using Cell Quest Pro software. A schematic plot was used to display the results: the lower left quadrant represents live cells; the lower right and upper right quadrants represent early and late apoptotic cells, respectively; the upper left quadrant represents necrotic cells. Apoptosis refers to the sum of early and late apoptotic cells.

### 2.5. Comet Assay for DNA Damage

DNA damage in individual cells was assessed using an alkaline comet assay, as previously described [29]. After respective treatments, cells were detached from culture surfaces, mixed with 0.75% low-melting agarose (LMA), and transferred to a microscope slide precoated with a layer of 0.75% normal-melting agarose. After solidification, slides were covered with LMA and immersed in lysis solution (2.5 M NaCl, 100 mM Na-ethylenediaminetetraacetic acid (EDTA), 10 mM Tris, 1% Triton X-100, and 10% DMSO (pH 10)) for 1 h at 4 °C to remove proteins. The slides were then placed in a horizontal electrophoresis tank containing electrophoresis buffer (300 mM NaOH, 10 mM Na-EDTA, pH 10) for 20 min to allow DNA unwinding under alkaline/neutral conditions. Thereafter, electrophoresis was conducted in the same buffer for 20 min at 4 °C to draw negatively-charged DNA towards the anode. Slides were then rinsed gently three times with neutralization buffer (0.4 M Tris-HCl, pH 7.5) for 10 min at 25 °C, stained with ethidium bromide (EtBr, 40 μg/mL, Sigma-Aldrich Chemical Co.), and observed under a fluorescence microscope at × 200 magnification. All of the steps above were conducted under yellow light to prevent DNA damage.

### 2.6. Determination of 8-Hydroxy-2′-deoxyguanosine (8-OHdG)

The BIOXYTECH^®^ 8-OHdG-EIA™ kit (OXIS Health Products Inc., Portland, OR, USA) was used to quantify oxidative DNA damage. Briefly, cellular DNA was isolated using a DNA Extraction Kit (iNtRON Biotechnology Inc., Sungnam, Republic of Korea) and quantified, according to the manufacturer’s protocol. The amount of 8-OHdG (a deoxyriboside form of 8-oxoguanine) in DNA was using a standard curve by measuring absorbance at 450 nm using a microplate reader (Dynatech Laboratories, Chantilly, VA, USA) according to the manufacturer’s instructions.

### 2.7. Measurement of ROS

Intracellular ROS levels were assessed by staining cells using the oxidation-sensitive dye 5,6-carboxy-2′,7′-dichlorofluorescein diacetate (DCF-DA). In brief, cells were trypsinized, suspended in PBS, and stained with DCF-DA (Sigma-Aldrich Chemical Co.) at a final concentration of 10 μM for 20 min at 37 °C in the dark. Relative fluorescence intensities of cell suspensions were measured using a flow cytometer. To assess intracellular ROS production by image analysis, cells were seeded on a coverslip loaded 6-well plate for 24 h, treated with nargenicin A1 or NAC for 1 h, and then with tacrolimus for 1 h. After washing with PBS, cells were stained with 10 μM DCF-DA for 20 min at 37 °C, washed, and mounted on microscope slides using mounting medium (Sigma-Aldrich Chemical Co.). The images were obtained using a fluorescence microscope at ×200 magnification.

### 2.8. Measurement of MMPs (Δψm)

To assess MMP losses, cells were collected and incubated in media containing 10μM of 5,5‘6,6’-tetrachloro-1,1’,3,3’-tetraethyl-imidacarbocyanine iodide (JC-1, Sigma-Aldrich Chemical Co.), a mitochondria-specific fluorescent dye, for 20min at 37 °C in the dark. After washing twice with PBS to remove unbound dye, green (JC-1 monomers) to red (JC-1 aggregates) fluorescence ratios were determined by a flow cytometer, according to the manufacturer’s instructions.

### 2.9. Determination of ATP Levels

Intracellular ATP levels were determined using a firefly-luciferase-based ATP Bioluminescence Assay Kit (Roche Applied Science, Indianapolis, IN, USA). Briefly, cells cultured under various conditions were lysed with the provided lysis buffer, and the collected supernatant was mixed with an equal amount of luciferase agent, which catalyzed the reaction between ATP and luciferin. Emitted light was immediately measured using a microplate luminometer and ATP levels were calculated using a standard curve. Intracellular ATP levels were expressed as percentages of untreated controls.

### 2.10. Western Blot Analysis

Cells were harvested, washed with PBS, and lysed with lysis buffer for 30 min, as previously described [30]. Protein concentrations were quantified using a Bio-Rad protein analysis kit (Bio-Rad Lab., Hercules, CA, USA). Equal amounts of proteins were subjected to sodium dodecyl sulfate-polyacrylamide gel electrophoresis and then transferred to polyvinylidene fluoride membranes (Millipore, Bedford, MA, USA), which were blocked with 5% bovine serum albumin (Sigma-Aldrich Chemical Co.) in a Tris-buffered saline/Tween-20 (TBST) and probed with primary antibodies overnight at 4 °C. Membranes were then incubated with appropriate secondary antibodies conjugated with horseradish peroxidase (HRP) for 2 h at room temperature and rinsed three times with TBST. Protein bands were visualized by incubating membranes in an enhanced chemiluminescence (ECL) reagent (Amersham Biosciences, Westborough, MA, USA), according to the manufacturer’s instructions. The immunoreactive bands were detected and exposed to X-ray film. Images of Western blotting were also analyzed using Image (Vilber Lourmat, Torcy, France).

### 2.11. Colorimetric Assay of Caspase-3 Activity

Caspase-3 activities were assessed using a colorimetric assay kit (R&D Systems, Minneapolis, MN, USA). Briefly, collected cells were lysed after treatments, and equal amounts of proteins were incubated with the supplied reaction buffer, which contained the caspase-3 substrate dithiothreitol and tetrapeptides (Asp-Glu-Val-Asp (DEAD)-p-nitroaniline (pNA)), for 2 h at 37 °C in the dark. Changes in absorbance at 405 nm were determined using a microplate luminometer according to the manufacturer’s instructions. Results are presented as multiples of untreated control cell values.

### 2.12. Statistical Analysis

All experiments were performed at least three times. Data were analyzed using GraphPad Prism software (version 5.03; GraphPad Software, Inc., La Jolla, CA, USA), and expressed as the mean ± standard deviation (SD). Differences between groups were assessed using analysis of variance followed by ANOVA-Tukey’s post hoc test, and *p* < 0.05 was considered to indicate a statistically significant difference.

## 3. Results

### 3.1. Nargenicin A1 Inhibited Tacrolimus-Induced Cytotoxicity in HINAE Cells

To establish experimental conditions, HINAE cells were treated with various concentrations of tacrolimus and/or nargenicin A1 for 24 h and then subjected to MTT assay. As shown in Figure 1A, tacrolimus concentration-dependently reduced cell viability. Nargenicin A1 was shown to reduce cells viability at concentrations above 25 μM but caused no significant change at 10 μM (Figure 1B). To evaluate the protective effect of nargenicin A1 on tacrolimus-induced cytotoxicity, nargenicin A1 was treated with 10 μM or less for 1 h and treated with 25 μM of tacrolimus for 24 h. As shown in Figure 1C, pretreatment with 1 μM nargenicin A1 did not affect the viability of HINAE cells treated with 25 μM tacrolimus; however, pretreatment with 10 or 50 μM of nargenicin A1 significantly and concentration-dependently protected cells from tacrolimus. In addition, NAC pretreatment completely suppressed the cytotoxicity of tacrolimus (Figure 1C).

### 3.2. Nargenicin A1 Suppressed Tacrolimus-Induced Apoptosis in HINAE Cells

To investigate whether the cytoprotective effect of nargenicin A1 was associated with apoptosis suppression, DAPI staining and flow cytometry were performed. Fluorescent images revealed that the control cells had intact nuclei, whereas tacrolimus-treated cells showed marked chromatin condensation, a characteristic of apoptosis (Figure 2A). However, tacrolimus-induced nuclear morphological changes were attenuated by nargenicin A1 or NAC pretreatment. Annexin V/PI double staining also showed nargenicin A1 pretreatment significantly reduced the apoptosis of tacrolimus-treated cells (Figure 2B,C).

### 3.3. Nargenicin A1 Reduced Tacrolimus-Induced DNA Damage in HINAE Cells

Comet assay was used to determine whether tacrolimus-induced apoptosis was associated with DNA damage, and whether nargenicin A1 prevented DNA damage. As shown in Figure 3A, the smeared nuclear DNA pattern was not observed in cells treated with nargenicin A1 or NAC or in control cells. However, tacrolimus-treated cells produced the smeared pattern, but the tail lengths produced by nargenicin A1 or NAC pretreated cells were obviously shorter. We also investigated the protective effect of nargenicin A1 on DNA damage by assessing the level of 8-OHdG, a specific marker of DNA oxidative damage. Tacrolimus increased 8-OHdG production by about 2.5-fold compared to the control, but nargenicin A1 significantly reduced tacrolimus-induced 8-OHdG production and NAC maintained 8-OHdG levels at the control level. These results suggest tacrolimus-induced apoptosis was related to DNA damage and that the protection afforded by nargenicin A1 against DNA damage suppressed apoptosis induction.

### 3.4. Nargenicin A1 Attenuated Tacrolimus-Induced ROS Generation in HINAE Cells

To determine whether the observed protective effects of nargenicin A1 were due to the suppression of oxidative stress, we examined the levels of ROS using a fluorescent probe, 5,6-carboxy-2′,7′-dichlorofluorescein diacetate (DCF-DA). As shown in Figure 4, intracellular fluorescence intensity was significantly increased within 1 h after tacrolimus exposure, but this increase was were significantly reduced by nargenicin A1 pretreatment and almost completely blocked by NAC pretreatment. Furthermore, treatment with nargenicin A1 alone caused no significant change in DCF-DA fluorescence intensities.

### 3.5. Nargenicin A1 Suppressed Tacrolimus-Induced Mitochondrial Dysfunction in HINAE cells

To investigate whether the ROS-mediated cytotoxic effect of tacrolimus was associated with mitochondrial dysfunction, we investigated MMPs and intracellular ATP levels. JC-1 staining results showed that tacrolimus increased the rate of depolarized cell population by about 2.4 times over the untreated controls (Figs. 5A and B). In addition, intracellular ATP concentrations in cells exposed to tacrolimus were significantly lower than those of the control group (Figure 5C). However, in the cells pretreated with nargenicin A1 or NAC, these changes by tacrolimus were significantly inhibited.

### 3.6. Nargenicin A1 Prevented Tacrolimus-Induced B-cell lymphoma-2 (Bcl-2)/Bax Ratio Reduction and the Activation of Caspase-3 in HINAE Cells

To investigate the mechanisms involved in apoptosis induced by tacrolimus, we examined the protein levels of Bcl-2 family members such as Bcl-2, Bax and caspase-3. Immunoblotting results showed that the protein expression levels of anti-apoptotic Bcl-2 and pro-apoptotic Bax were significantly inhibited and increased by tacrolimus, respectively. However, pretreatment with nargenicin A1 or NAC significantly inhibited these changes (Figure 6A,B). Furthermore, tacrolimus treatment significantly increased caspase-3 activation, which was also attenuated by nargenicin A1 or NAC pretreatment (Figure 6C).

## 4. Discussion

Although immunosuppressants are widely used to treat many diseases, long-term treatment may have serious adverse effects, and thus, new strategies are required [31]. In many previous studies, tacrolimus has been shown to cause oxidative stress in various tissues [32,33]. For example, intracellular ROS accumulation has been shown to play a key role in the apoptosis of kidney and renal cells by tacrolimus [16,34], and oxidative stress caused by excessive ROS production also plays a major role in the induction of apoptosis in Jurkat human T lymphocytes [35,36]. Additionally, tacrolimus is known to have a close relationship to side effects in transplant recipients; specifically, the whole blood concentrations of tacrolimus at 20–30 ng mL^−1^ provoked transplant rejection in 41% of recipients [37]. Therefore, in the current study, we evaluated the efficacy of nargenicin A1 as an inhibitor of cytotoxicity by tacrolimus as a screening program for the development of new immunostimulants in fish culture under tacrolimus (25 μM = 21 mg mL^−1^) concentration in vivo. In this study, we evaluated the efficacy of nargenicin A1 as an inhibitor of cytotoxicity by tacrolimus as a screening program for the development of new immunostimulants in fish culture. Our results demonstrate that nargenicin A1 pretreatment effectively prevented tacrolimus-induced DNA damage and apoptosis by reducing ROS accumulation, which suggests tacrolimus-induced cytotoxicity and apoptosis were caused by ROS-mediated DNA damage. These results agree well with those of previous studies, which showed that the inhibition of tacrolimus-induced apoptosis by taurine or green tea polyphenols was associated with the suppression of ROS production [16,34], suggesting that the antioxidant capacity of nargenicin A1 is primarily responsible for reducing tacrolimus-induced ROS generation and oxidative stress.

ROS overload also results in free radical attack of membrane phospholipids and in turn leads to mitochondrial membrane depolarization and loss of MMP, which is considered the first step of the mitochondria-mediated intrinsic apoptosis pathway [19,38]. Mitochondrial dysfunction also causes abnormalities in the electron transport pathways of the mitochondrial respiratory chain and ultimately disrupts intracellular ATP production [20,39]. In the present study, MMP and ATP levels were markedly reduced when HINAE cells were exposed to tacrolimus, and pretreatment with nargenicin A1 or NAC significantly suppressed this tacrolimus-induced loss of MMP and APT, which indicates nargenicin A1 protected cells against oxidative stress-induced mitochondrial dysfunction.

The activation of mitochondria-mediated apoptosis is regulated by various proteins, including members of the Bcl-2 family, which contains anti- and pro-apoptotic proteins. Anti-apoptotic proteins like Bcl-2 are located on the outer mitochondrial membrane and prevent the release of apoptogenic factors [40,41]. On the other hand, pro-apoptotic proteins, like Bax, antagonize the activities of anti-apoptotic proteins or translocate to the mitochondrial membrane to form membrane-integrated homo-oligomers that cause mitochondrial pore formation and loss of MMP, and subsequent release of apoptotic factors from cytosol [41,42]. Therefore, the balance between apoptotic Bax proteins and anti-apoptotic Bcl-2 proteins determines activation and inhibition of the intrinsic apoptosis pathway. Consistent with the results of this study, previous studies that used different experimental models have reported tacrolimus-induced apoptosis was associated with a decrease in Bcl-2/Bax ratio and/or caspase-3 activation [16,35,43,44,45,46]. Moreover, several natural products with antioxidant properties have been shown to protect against tacrolimus-mediated cytotoxicity and tacrolimus-induced suppressions of Bcl-2/Bax ratio and/or caspase-3 activation [16,34,43,44], which are very similar to the results of this study. Accordingly, the results of this study suggest that ROS-dependent mitochondrial dysfunction plays a pivotal role in the induction of apoptosis by tacrolimus in HINAE cells, and the blockade of cytotoxicity by nargenicin A1 is due to the antioxidant ability of this antibiotic.

## 5. Conclusions

In summary, this is the first study to show nargenicin A1 effectively prevents tacrolimus-induced oxidative stress, mitochondrial dysfunction, DNA damage, and apoptosis. In addition, nargenicin A1 suppressed tacrolimus-induced Bcl-2/Bax ratio reduction and caspase-3 activation. We suggest additional studies, including in vivo studies, be conducted to determine whether pathways other than the intrinsic pathway are involved tacrolimus-induced apoptosis.

## Figures and Tables

**Figure 1 ijerph-16-01044-f001:**
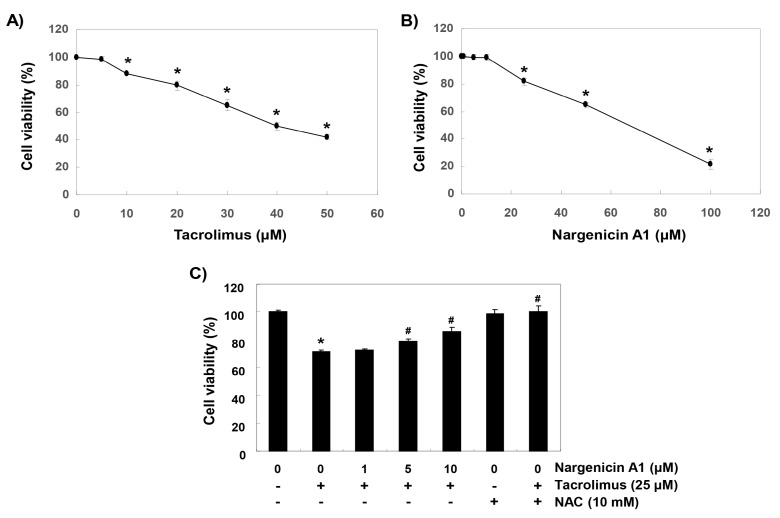
Protective effects of nargenicin A1 against tacrolimus-induced cytotoxicity in hirame natural embryo (HINAE) cells. Cells were treated with various concentrations of tacrolimus or nargenicin A1 for 24 h (**A**,**B**) or pretreated with nargenicin A1 or *N*-acetyl cysteine (NAC) at the indicated concentrations for 1 h and then treated with 25 μM of tacrolimus for 24 h (**C**). After treatments, cell viabilities were assessed using 3-(4,5-dimethylthiazol-2-yl)-2,5-diphenyltetrazolium bromide (MTT) assay. Results are presented as the means ± SD of three independent experiments (* *p* < 0.05 vs. non-treated controls, ^#^
*p* < 0.05 versus tacrolimus treated cells).

**Figure 2 ijerph-16-01044-f002:**
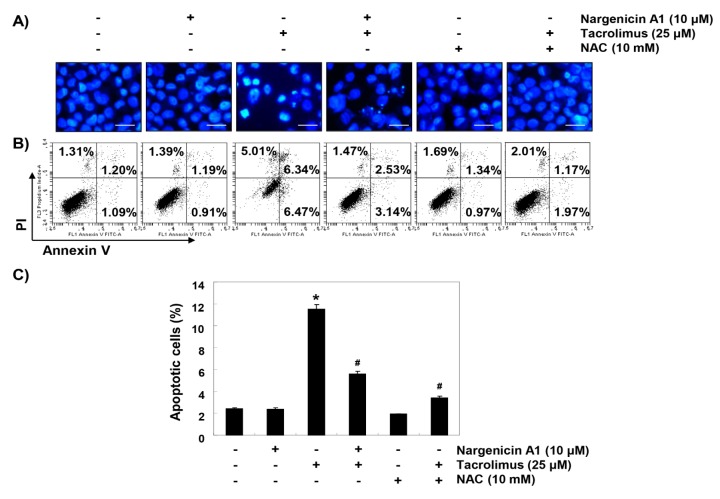
Suppression of tacrolimus-induced apoptosis by nargenicin A1 in HINAE cells. HINAE cells were treated with 10 μM nargenicin A1 or 10 mM NAC for 1 h, and then stimulated with or without 25 μM tacrolimus for 24 h. (**A**) Cells were collected, fixed, stained with 4,6-diamidino-2-phenylindole (DAPI), and photographed under a fluorescence microscope (original magnification, ×400). Scale bar, 50 µm. (**B**,**C**) Cells cultured under the same conditions were collected and stained with fluorescein isothiocyanate (FITC)-conjugated annexin V and propidium iodide (PI) for flow cytometry. (**B**) Results showed necrosis, defined as annexin V-negative and PI-positive cells (lower upper quadrant); early apoptosis, defined as annexin V-positive and PI-negative cells (lower right quadrant); and late apoptosis, defined as annexin V-positive and PI-positive (upper right quadrant) cells. (**C**) Percentages of apoptotic cells were determined by expressing the number of annexin V-positive cells as a percentage of all cells present. Results are presented as the means ± SD of three independent experiments (* *p* < 0.05 versus non-treated controls, # *p* < 0.05 versus tacrolimus-treated cells).

**Figure 3 ijerph-16-01044-f003:**
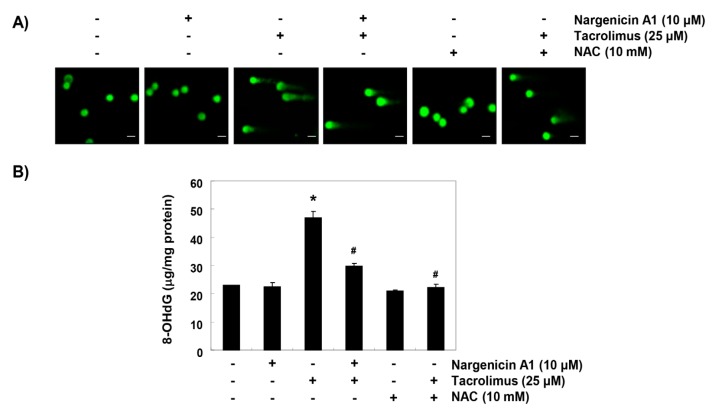
Protection effect of nargenicin A1 on tacrolimus-induced DNA damage in HINAE cells. (**A**) Cells were pretreated with 10 µM nargenicin A1 or 10 mM NAC for 1 h, and then stimulated with or without 25 μM tacrolimus for 24 h. Comet assays were used to detect DNA damage. The representative images shown were taken using a fluorescence microscope (original magnification, ×200). Scale bar, 20 µm. (**B**) Cellular DNA was isolated from cells grown under the same conditions and quantified using a spectrophotometer. Amounts of 8-OHdG in DNA were determined using an 8-OHdG-EIA kit. Results are presented as the means ± SD of triplicate measurements (* *p* < 0.05 versus non-treated controls, ^#^
*p* < 0.05 versus tacrolimus treated cells).

**Figure 4 ijerph-16-01044-f004:**
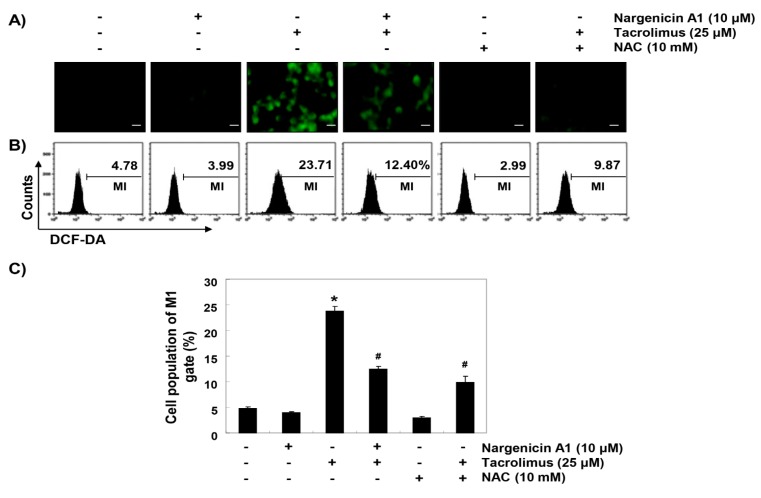
Suppression of tacrolimus-induced reactive oxygen species (ROS) accumulation by nargenicin A1 in HINAE cells. Cells were pretreated with 10 µM nargenicin A1 or 10 mM NAC for 1 h, and then stimulated with or without 25 μM tacrolimus for 1 h. (**A**) After staining with 5,6-carboxy-2′,7′-dichlorofluorescein diacetate (DCF-DA), images were obtained using a fluorescence microscope (original magnification, ×200). Scale bar, 20 µm. (**B**) DCF fluorescence was monitored using a flow cytometer. (**C**) Results are presented as the means ± SD of three independent experiments (* *p* < 0.05 versus non-treated controls, ^#^
*p* < 0.05 versus tacrolimus-treated cells).

**Figure 5 ijerph-16-01044-f005:**
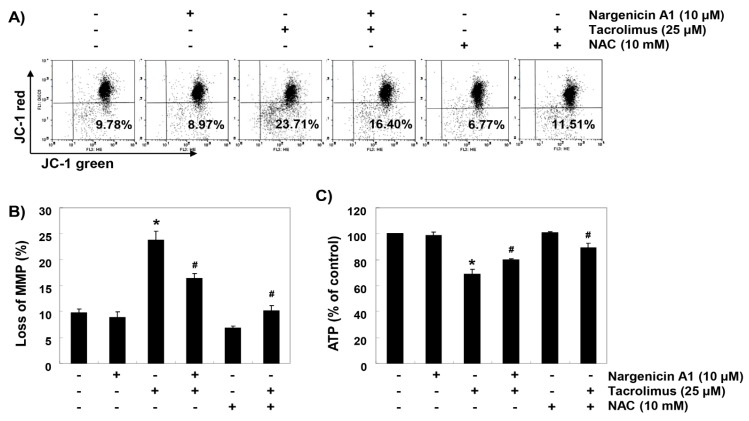
Attenuation of tacrolimus-induced mitochondrial dysfunction by nargenicin A1 in HINAE cells. Cells were pretreated with 10 µM nargenicin A1 or 10 mM NAC for 1 h and then stimulated with or without 25 μM tacrolimus for 24 h. (**A**) Cells were collected and incubated with 10 µM JC-1, and mitochondrial membrane potential (MMP) values were determined by flow cytometry. (**B**) Results are presented as the means ± SD of three independent experiments (* *p* < 0.05 versus non-treated controls, ^#^
*p* < 0.05 versus tacrolimus-treated cells). (**C**) A commercial kit was used to monitor ATP production. Results are presented as the means ± SD of three independent experiments (* *p* < 0.05 versus non-treated controls, ^#^
*p* < 0.05 versus tacrolimus-treated cells).

**Figure 6 ijerph-16-01044-f006:**
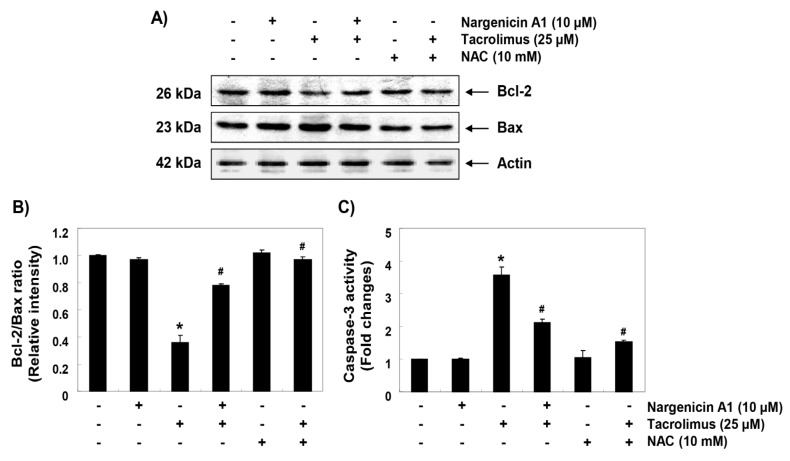
Effect of nargenicin A1 on the expressions of Bcl-2 and Bax and on the activity of caspase-3 in tacrolimus-treated HINAE cells. (**A**) Cells were co-treated with 10 µM nargenicin A1 or 10 mM NAC with or without 25 μM tacrolimus for 24 h. Bcl-2 and Bax protein levels were assayed by Western blotting using an enhanced chemiluminescence (ECL) detection system using actin as the internal control. (**B**) Bands were quantified using ImageJ, normalized to actin and ratios were determined. (**C**) Cells lysates were assayed for caspase-3 activity using DEAD-pNA as substrate. Amounts of pNA released were determined by measuring at 405 nm using an ELISA microplate reader. Results are presented as the means ± SD of three independent experiments (* *p* < 0.05 versus non-treated controls, ^#^
*p* < 0.05 versus tacrolimus-treated cells).

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
