# Peer review of "Protective Effects of Nargenicin A1 against Tacrolimus-Induced Oxidative Stress in Hirame Natural Embryo Cells"

_ijerph, 2019, doi:10.3390/ijerph16061044_

Round 1

Reviewer 1 Report

The Authors compared the bioactivity of two antibiotics Tacrolimus and Nargenicin A1 using the model of HINAE cell line. The main thesis of the paper is to determine the protective effects of nargenicin A1 on DNA damage and apoptosis induced by tacrolimus. I think that the idea of this research is interesting, but the presentation of the data diminishes the impact of the paper.

The authors determined cells viability using MTT assay as well as analysed apoptosis based on the morphology of nuclei and Annexin V/PI staining. Further, Authors performed comet assay to determine the DNA damage at experimental conditions and found that tacrolimus-induced  apoptosis  is associated  with  DNA  damage  and  that  the  protection  afforded  by  nargenicin  A1  against  DNA  damage might suppress apoptosis induction. The Authors also determined the mitochondrial potential of the cells, ROS activity and detected typical markers of apoptosis.

To conclude the methods were correctly planned and performed in order to proof the thesis, however, I have concerns in terms of description of methodology and presentation of data:

1.      The Authors should explain why they decided to perform this studies using model of hirame natural embryo cells? It should be somehow justify in the introduction section or discussion.

2.      Based on Annexin staining authors should comment the influence of tested compounds on apopotosis profile (early and late apoptotic cells).

Additionally, Figure 2 A should be supplemented with the proper scale bar.

3.      The quality of ROS images is very poor. Please provide more representative images, also include scale bar.

4.      For Western blot membranes – the molecular mass marker should be provided or at least the molecular weight of proteins should be indicted.

5.      Figure 5B refers to the densitometric analysis of WB bands, please provide information which software was used for this analysis in the materials sections.

6.      Further, Figure 5B is cited as a figure showing Caspase-3 activity – please provide proper figure showing results of ELISA measurement.

Author Response

Response to Reviewer 1 Comments

The Authors compared the bioactivity of two antibiotics Tacrolimus and Nargenicin A1 using the model of HINAE cell line. The main thesis of the paper is to determine the protective effects of nargenicin A1 on DNA damage and apoptosis induced by tacrolimus. I think that the idea of this research is interesting, but the presentation of the data diminishes the impact of the paper.

The authors determined cells viability using MTT assay as well as analysed apoptosis based on the morphology of nuclei and Annexin V/PI staining. Further, Authors performed comet assay to determine the DNA damage at experimental conditions and found that tacrolimus-induced apoptosis is associated with DNA damage and that the protection afforded by nargenicin A1 against DNA damage might suppress apoptosis induction. The Authors also determined the mitochondrial potential of the cells, ROS activity and detected typical markers of apoptosis.

To conclude the methods were correctly planned and performed in order to proof the thesis, however, I have concerns in terms of description of methodology and presentation of data:

1. The Authors should explain why they decided to perform this studies using model of hirame natural embryo cells? It should be somehow justify in the introduction section or discussion.

Response 1: The most problematic aspect of the aquaculture of fish is the high mortality rate of young fish due to various causes. Therefore, it is urgent to develop an immune-enhancing agent that can help immune enhancement of young fish. In this study, we investigated the possibility of using nargenicin A1 as an immunity enhancer and selected tacrolimus as an immunosuppressive agent. I do not think our results can be used immediately in the field. However, based on these studies, it would be meaningful to investigate whether there is any possibility of developing nargenicin A1 or derivatives of this substance with immunity enhancing effect. Therefore, please understand that this study has been tried as part of screening program for the development of new immunostimulants.

2. Based on Annexin staining authors should comment the influence of tested compounds on apopotosis profile (early and late apoptotic cells).

Response 2: According to your opinion, the manuscripts of the relevant parts have been revised.

Additionally, Figure 2 A should be supplemented with the proper scale bar.

Response: According to your comments, we have added the scale bars in each Figure and information about magnification in the Figures legends.

3. The quality of ROS images is very poor. Please provide more representative images, also include scale bar.

Response 3: According to your comments, the images of the ROS were replaced and scale bars were also added.

4. For Western blot membranes – the molecular mass marker should be provided or at least the molecular weight of proteins should be indicted.

Response 4: According to your suggestion, we have presented the molecular weights for each protein.

5. Figure 5B refers to the densitometric analysis of WB bands, please provide information which software was used for this analysis in the materials sections.

Response 5: We have added information about the software used for densitometric analysis of Western blot results according to your recommendation.

6. Further, Figure 5B is cited as a figure showing Caspase-3 activity – please provide proper figure showing results of ELISA measurement.

Response 6: This was an error during the writing of the paper and we have replaced Figure 5B, which contains the results of Caspase-3 activity. Thank you for pointing out the error.

We are appreciated for your comments and suggestions on our manuscript to improve our study.

Reviewer 2 Report

The present manuscript focuses on an important topic of Pharmacology / Toxicology / Public Health: the toxicity induced by some chemotherapeutic and immunosuppressant agents and the ability to mitigate it.

Regarding the English language and style, extensive editing of the English language and style are required. There are language and grammatical errors and incorrect scientific terms or expressions. I highlighted some examples throughout the manuscript. I also included some comments directly on the document. Additionally, the authors often use too long sentences that make it difficult to understand the meaning of their statements (example: Page 2 / Lines 50-54).

I have several doubts and concerns throughout this manuscript, namely regarding the methodologies used. There is important information missing or raises some questions.

I summarized some of the topics that should be improved, justified or clarified:

1. Introduction

In my opinion, besides some grammatical issues, some topics should be more detailed (e.g. previous studies developed with Nargenicin A1)

2. Methods

2.1. Cell culture and drug treatment

I) It is not clear why the authors used HINAE cells. In my opinion, it would be essential to perform this study in a human cell line.

II) The authors should explain why the cells were incubated at 20ºC and without any % of CO2.

III)   Tacrolimus and nargenicin A1 were prepared in DMSO with subsequent dilutions with culture medium. What was the final concentration of DMSO per well?

2.2.  Cell viability assays

How many internal replicates were performed in each plate/independent assay? In my opinion, this information should be present in this section.

2.3.  Apoptosis assay and 2.7. Measurement of ROS

The authors should include the final magnification.

2.7. Measurement of ROS

I) How were the cells detached prior to the assay (flow cytometry)? What is the reagent used?

The use of the fluorescence probe DCF presents several drawbacks, as discussed in the oxidative stress field in the last years. The intracellular redox chemistry of this probe is complex and has several limitations and could be affected by several artifacts.

Thus, the authors should have used other fluorescence probe.

2.12. Statistical analysis

In the manuscript is not clear what type of ANOVA and post-hoc tests were used.

3. Results

I) Figure 1: The graphics A and B should be “scatter XY plot” to better visualize the concentration / toxic profile of each compound and not a “bar graph”.

The legend has unnecessary information (number of cells seed for 24h) and the statistical test is missing.

II) What is the clinical relevance of the concentration of tacrolimus herein tested (25 µM)?

III) Regarding the apoptosis assay, it was not mentioned the % of necrosis (PI staining). In Figure 2B, it is possible to observe a significant increase not only in the apoptosis but also in the necrosis (PI+ cells) that should be mentioned.

IV) In the following sections of the “Results” there are similar errors or information missing that should be corrected by the authors.

V)  In the 3.4. Section / Figure 4 it is observed a different shape of the curve (of ROS) for each condition, probability due to some different ROS produced (it is more evident for tacrolimus-exposed cells). For this reason, would be preferable to measure the increase of ROS using the median of the curves/graphs (a widely used measurement for this type of assay), instead of the % that the authors used (right section of the curves).

Author Response

Response to Reviewer 2 Comments

The present manuscript focuses on an important topic of Pharmacology / Toxicology / Public Health: the toxicity induced by some chemotherapeutic and immunosuppressant agents and the ability to mitigate it.

Regarding the English language and style, extensive editing of the English language and style are required. There are language and grammatical errors and incorrect scientific terms or expressions. I highlighted some examples throughout the manuscript. I also included some comments directly on the document. Additionally, the authors often use too long sentences that make it difficult to understand the meaning of their statements (example: Page 2 / Lines 50-54).

: Thank you for your kind suggestion. Based on your suggestion, we have modified a number of sentences, including the corresponding sentence.

I have several doubts and concerns throughout this manuscript, namely regarding the methodologies used. There is important information missing or raises some questions.

I summarized some of the topics that should be improved, justified or clarified:

Response: We are appreciated for your suggestions on our manuscript. According to your comments, the manuscript has been revised as follows.

1. Introduction

In my opinion, besides some grammatical issues, some topics should be more detailed (e.g. previous studies developed with Nargenicin A1)

Response: Thank you for your kind suggestion. According to your suggestion, we modified some parts of the Introduction.

2. Methods

2.1. Cell culture and drug treatment

I) It is not clear why the authors used HINAE cells. In my opinion, it would be essential to perform this study in a human cell line.

Response: We fully agree with the suggestion you indicated. Currently, we are conducting experiments on human-derived cells for other projects.

The most problematic aspect of the current fish culture is the high mortality rate of young fish due to various causes. Therefore, it is urgent to develop an immune-enhancing agent that can help immune enhancement of young fish. In this study, we investigated the possibility of using nargenicin A1 as an immunity enhancer and selected tacrolimus as an immunosuppressive agent. I do not think our results can be used immediately in the field. However, based on these studies, it would be meaningful to investigate whether there is any possibility of developing nargenicin A1 or derivatives of this substance with immunity enhancing effect. Therefore, please understand that this study has been tried as part of screening program for the development of new immunostimulants.

The results of this study will be useful as comparative data for human-derived cells (hepatocytes) currently being performed.

II) The authors should explain why the cells were incubated at 20ºC and without any % of CO2.

Response: Culture of HINAE cells for this experiment was carried out according to the previous study method, and there was no problem with cell culture under these conditions.

References:

Kasai H, Yoshimizu M. Establishment of two Japanese flounder embryo cell lines. Bull. Fish Sci. Hokkaido Univ. 2001, 52, 67-70.

Ohtani M, Hikima J, Kondo H, Hirono I, Jung TS, Aoki T. Characterization and antiviral function of a cytosolic sensor gene, MDA5, in Japanese flounder, Paralichthys olivaceus. Dev Comp Immunol. 2011, 35, 554-562.

Xu H, Xing J, Tang X, Sheng X, Zhan W. Intramuscular administration of a DNA vaccine encoding OmpK antigen induces humoral and cellular immune responses in flounder (Paralichthys olivaceus) and improves protection against Vibrio anguillarum. Fish Shellfish Immunol. 2019, 86, 618-626.

III) Tacrolimus and nargenicin A1 were prepared in DMSO with subsequent dilutions with culture medium. What was the final concentration of DMSO per well?

Response: Thank you for your comments. The concentration of DMSO used in this study was less than 0.05%, which does not show cytotoxicity. This was added to the Methods and Materials section.

2.2. Cell viability assays

How many internal replicates were performed in each plate/independent assay? In my opinion, this information should be present in this section.

Response: According to your comments, I added the following sentence.

→ The experiments were repeated three times with at least triplicate wells for each concentration, and results are expressed as percentage reductions in absorbance versus non-treated controls.

2.3. Apoptosis assay and 2.7. Measurement of ROS

The authors should include the final magnification.

Response: According to your comments, the magnifications were added to the corresponding parts in the Materials and Methods and scale bars were also added in Figure 2A, Figure 3A and Figure 4A.

2.7. Measurement of ROS

I) How were the cells detached prior to the assay (flow cytometry)? What is the reagent used?

Response: We used trypsin and have provided this in the ''2.7. Measurement of ROS" section.

The use of the fluorescence probe DCF presents several drawbacks, as discussed in the oxidative stress field in the last years. The intracellular redox chemistry of this probe is complex and has several limitations and could be affected by several artifacts.

Thus, the authors should have used other fluorescence probe.

Response: We totally agree with what you pointed out. However, many researchers still use DCF-DA. We will use a more appropriate fluorescent probe in the future to rule out the drawbacks of DCF-DA.

2.12. Statistical analysis

In the manuscript is not clear what type of ANOVA and post-hoc tests were used.

Response: According to your comments, the details of the significance analysis used in this study are presented in detail in the "Statistical analysis" Section.

3. Results

I) Figure 1: The graphics A and B should be “scatter XY plot” to better visualize the concentration / toxic profile of each compound and not a “bar graph”.

The legend has unnecessary information (number of cells seed for 24h) and the statistical test is missing.

Response: Thank you for your kind suggestion. However, the graphs A and B show the average value of the results for the triplicate experiment, which is based on the method proposed by many researchers. Please take this into consideration. And, according to your comments, we removed the unnecessary information and presented the significance of the results to the legend.

II) What is the clinical relevance of the concentration of tacrolimus herein tested (25 µM)?

Response: The concentration of tacrolimus tested here (25 μM) is not clinically relevant. We selected this concentration because tacrolimus induces adequate cytotoxicity, ie, 60% survival. Clinical studies have shown that blood concentrations below 15 ng/ml should be maintained to minimize toxicity of tacrolimus.

III) Regarding the apoptosis assay, it was not mentioned the % of necrosis (PI staining). In Figure 2B, it is possible to observe a significant increase not only in the apoptosis but also in the necrosis (PI+ cells) that should be mentioned.

Response: According to your suggestion, the following sentence was added to the "2.4. Apoptosis analysis using a flow cytometer" section in the Materials and Methods.

→ A schematic plot was used to display the results: the lower left quadrant represents live cells; the lower right and upper right quadrants represent early and late apoptotic cells, respectively; the upper left quadrant represents necrotic cells. Apoptosis refers to the sum of early and late apoptotic cells.

: We also added the following sentence to the legend of the Figure 2 and corrected the corresponding result.

→ Results showed necrosis, defined as annexin V-negative and PI-positive cells (lower upper quadrant), early apoptosis, defined as annexin V-positive and PI-negative cells (lower right quadrant), and late apoptosis, defined as annexin V-positive and PI-positive (upper right quadrant) cells.

IV) In the following sections of the “Results” there are similar errors or information missing that should be corrected by the authors.

Response: According to your suggestion, we have modified many of the sentences in the Result section.

V) In the 3.4. Section / Figure 4 it is observed a different shape of the curve (of ROS) for each condition, probability due to some different ROS produced (it is more evident for tacrolimus-exposed cells). For this reason, would be preferable to measure the increase of ROS using the median of the curves/graphs (a widely used measurement for this type of assay), instead of the % that the authors used (right section of the curves).

Response: Thank you for your comment. As you well know, the X axis in Figure 4B represents the population (or fluorescence intensity) of cells belonging to M1 in flow cytometry results by DCF-DA staining. Therefore, we know that the X axis is expressed as a magnification of the relative fluorescence intensity or as a cell population of M1 gate (%).

We modified the X-axis to "Cell population of M1 gate (%)" instead of "Relative ROS levels (Fold changes)" and re-presented it.

Thank you for encouraging us to improve our study.

Round 2

Reviewer 1 Report

The manuscript was significantly improved. The Authors answered to all my concerns.

Author Response

Response to Reviewer 1 Comments

The manuscript was significantly improved. The Authors answered to all my concerns.

Response: Thanks to the positive evaluation and the careful comments on our manuscript.

Reviewer 2 Report

The manuscript was extensively revised by the authors and some of the questions and topics were addressed.

Nevertheless, there still some aspects that can be improved:

1)     Figure 1. The concentration-response profile should be performed in a XY scatter graph. When we look to the effect of a substance that can be dependent of the concentration we should not only look to “the bars” that have the same distance between them, even for different range of concentrations.

2)     Because “The concentration of tacrolimus tested here (25 μM) is not clinically relevant.”, the authors must include a sentence in the discussion section.

Overall, this study has some experimental limitations (the selection of the cell line, the use of trypsin before the measurement of ROS and the use of the DCF-DA fluorescence probe), which may lead to incomplete or incorrect conclusions.

Nevertheless, the article can be considered for publication since it may stimulate further studies on this subject.

Author Response

Response to Reviewer 2 Comments

The manuscript was extensively revised by the authors and some of the questions and topics were addressed.

Nevertheless, there still some aspects that can be improved:

1) Figure 1. The concentration-response profile should be performed in a XY scatter graph. When we look to the effect of a substance that can be dependent of the concentration we should not only look to “the bars” that have the same distance between them, even for different range of concentrations.

Response: According to your comments, the Figure 1 has been revised

2) Because “The concentration of tacrolimus tested here (25 μM) is not clinically relevant.”, the authors must include a sentence in the discussion section.

Response: Thank you for your comments. We partially corrected the "Discussion" section according to the comments you indicated.

In addition, we have provided a relevant reference.

Overall, this study has some experimental limitations (the selection of the cell line, the use of trypsin before the measurement of ROS and the use of the DCF-DA fluorescence probe), which may lead to incomplete or incorrect conclusions.

Response: We would like to inform you that your point of view has been a great help to us.

In the future, we will consider these issues carefully when doing similar experiments.

Nevertheless, the article can be considered for publication since it may stimulate further studies on this subject.

: Thank you for encouraging us to improve our study.
